# SDF-SRN: Learning Signed Distance 3D Object Reconstruction from Static Images

**Chen-Hsuan Lin**    **Chaoyang Wang**    **Simon Lucey**
Carnegie Mellon University
chlin@cmu.edu    chaoyanw@andrew.cmu.edu    slucey@cs.cmu.edu
https://chenhsuanlin.bitbucket.io/signed-distance-SRN/

## Abstract

Dense 3D object reconstruction from a single image has recently witnessed remarkable advances, but supervising neural networks with ground-truth 3D shapes is impractical due to the laborious process of creating paired image-shape datasets. Recent efforts have turned to learning 3D reconstruction without 3D supervision from RGB images with annotated 2D silhouettes, dramatically reducing the cost and effort of annotation. These techniques, however, remain impractical as they still require multi-view annotations of the same object instance during training. As a result, most experimental efforts to date have been limited to synthetic datasets. In this paper, we address this issue and propose SDF-SRN, an approach that requires only a *single view* of objects at *training time*, offering greater utility for real-world scenarios. SDF-SRN learns implicit 3D shape representations to handle arbitrary shape topologies that may exist in the datasets. To this end, we derive a novel differentiable rendering formulation for learning signed distance functions (SDF) from 2D silhouettes. Our method outperforms the state of the art under challenging single-view supervision settings on both synthetic and real-world datasets.

## 1   Introduction

Humans have strong capabilities to reason about 3D geometry in our visual world. When we see an object, not only can we infer its shape and appearance, but we can also speculate the underlying 3D structure. We learn to develop the concepts of 3D geometry and semantic priors, as well as the ability to mentally reconstruct the 3D world. Somehow through visual perception, *i.e.* just looking at a collection of 2D images, we have the ability to infer the 3D geometry of the objects in those images.

Researchers have sought to emulate such ability of 3D shape recovery from a single 2D image for AI systems, where success has been drawn specifically through neural networks. Although one could train such networks naively from images with associated ground-truth 3D shapes, such paired data are difficult to come by at scale. While most works have resorted to 3D object datasets, in which case synthetic image data can be created pain-free through rendering engines, the domain gap between synthetic and real images has prevented them from practical use. An abundant source of supervision that can be practically obtained for real-world image data is one problem that looms large in the field.

In the quest to eliminate the need for direct 3D supervision, recent research have attempted to tackle the problem of learning 3D shape recovery from 2D images with object silhouettes, which are easier to annotate in practice. This line of works seeks to maximize the reprojection consistency of 3D shape predictions to an ensemble of training images. While success has been shown on volumetric [41] and mesh-based [18, 25] reconstruction, such discretized 3D representations have drawbacks. Voxels are inefficient for representing shape surfaces as they are sparse by nature, while meshes are limited to deforming from fixed templates as learning adaptive mesh topologies is a nontrivial problem. Implicit shape representations become a more desirable choice for overcoming these limitations.

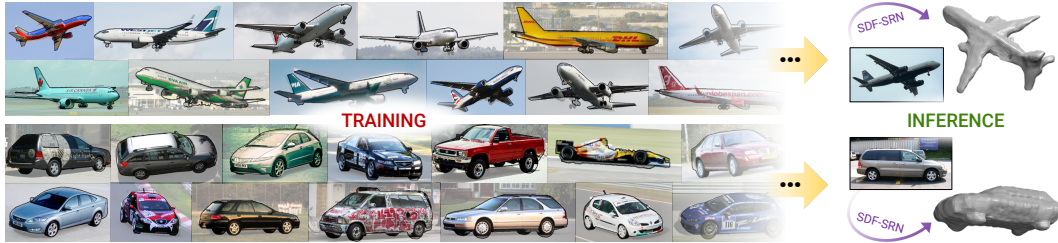

Figure 1: **Learning 3D SDF shape reconstruction from static images.** SDF-SRN learns implicit shape reconstruction from single-view images and 2D silhouettes at *training* time, allowing practical applications of real-world 3D object reconstruction trained from static image datasets.

Differentiable rendering methods for reconstructing 3D implicit representations have since sparked wide interest [27, 34]. Previous works, however, have required a multi-view setup, where objects are observed from multiple viewpoints *with* silhouette annotations. Since such data is difficult to obtain en masse, it has been unclear to the community how one can learn dense 3D reconstruction from single images at *training* time, where each individual object instance is assumed observed only once.

In this paper, we make significant advances on learning dense 3D object reconstruction from single images and silhouettes, *without* the knowledge of the underlying shape structure or topology. To this end, we derive a formulation to learn signed distance functions (SDF) as the implicit 3D representation from images, where we take advantage of distance transform on silhouettes to provide rich geometric supervision from all pixels of an image. In addition, we build a differentiable rendering framework upon the recently proposed Scene Representation Network [39] for efficient optimization of shape surfaces. The proposed method, SDF-SRN, achieves state-of-the-art 3D object reconstruction results on challenging scenarios that requires only a single observation for each instance during *training* time. SDF-SRN also learns high-quality 3D reconstruction from real-world static images with *single-view* supervision (Fig. 1), which was not possible with previous implicit shape reconstruction methods.

In summary, we present the following contributions:

- We establish a novel mathematical formulation to optimize 3D SDF representations from 2D distance transform maps for learning dense 3D object reconstruction without 3D supervision.
- We propose an extended differentiable rendering algorithm that efficiently optimizes for the 3D shape surfaces from RGB images, which we show to be suitable only for SDF representations.
- Our method, SDF-SRN, significantly outperforms state-of-the-art 3D reconstruction methods on ShapeNet [3] trained with *single*-view supervision, as well as natural images from the PASCAL3D+ [45] dataset *without* 3D supervision nor externally pretrained 2.5D/3D priors.

## 2    Related Work

**Learning 3D reconstruction without 3D supervision.** The classical shape-from-silhouette problem [21, 17] can be posed as a learning problem by supervising the visual hull with a collection of silhouette images using a differentiable projection module. The reconstructed visual hull can take the forms of voxels [46, 8], point clouds [23], or 3D meshes [18, 25]. RGB images can also serve as additional supervisory signals if available [41, 16, 24]. While these methods learn from 2D supervision sources that are easier to obtain, they typically require multi-view observations of the same objects to be available. To this end, Kanazawa *et al.* [15] took a first step towards 3D reconstruction from static image collections by jointly optimizing for mesh textures and shape deformations.

Deep implicit 3D representations [4, 31, 37] has recently attracted wide interest to 3D reconstruction problems for their power to model complex shape topologies at arbitrary resolutions. Research efforts on learning implicit 3D shapes without 3D supervision have primarily resorted to binary occupancy [26, 34] as the representation, aiming to match reprojected 3D occupancy to the given binary masks. Current works adopting signed distance functions (SDF) either require a pretrained deep shape prior [27] or are limited to discretized representations [14] that do not scale up with resolution. Our method learns deep SDF representations *without* pretrained priors by establishing a more explicit geometric connection to 2D silhouettes via distance transform functions.

**Neural image rendering.** Rendering 2D images from 3D shapes is classically a non-differentiable operation in computer graphics, but recent research has advanced on making the operation differentiable and incorporable with neural networks. Differentiable (neural) rendering has been utilized for learning implicit 3D-aware representations [28, 38, 33], where earlier methods encode 3D voxelized features respecting the corresponding 2D pixel locations of images. Such 3D-aware representations, however, refrain one from interpreting *explicit* 3D geometric structures that underlies in the images. Recently, Scene Representation Networks (SRN) [39] offered up a solution of learning depth from images by keeping a close proximity of neural rendering to classical ray-tracing in computer graphics. An advantage of SRN lies in its efficiency by *learning* the ray-tracing steps, in contrast to methods that requires dense sampling along the rays [34, 32]. We take advantage of this property for 3D reconstruction, which we distinguish from "3D-aware representations", in the sense that globally consistent 3D geometric structures are recovered instead of view-dependent depth predictions.

## 3 Approach

Our 3D shape representation is a continuous implicit function $f : \mathbb{R}^3 \to \mathbb{R}$, where the 3D surface is defined by the zero level set $\mathcal{S} = \{\mathbf{x} \in \mathbb{R}^3 \mid f(\mathbf{x}) = 0\}$. We define $f$ as a multi-layer perceptron (MLP); since MLPs are composed of continuous functions (*i.e.* linear transformations and nonlinear activations), $f$ is also continuous (almost everywhere) by construction. Therefore, the surface of a 3D shape defined by $\mathcal{S}$ is dense and continuous, forming a 2-manifold embedded in the 3D space.

### 3.1 Learning 3D Signed Distance Functions from 2D Silhouettes

Shape silhouettes are important for learning 3D object representations: the projection of a 3D shape should match the silhouettes of a 2D observation under the given camera pose. One straightforward approach is to utilize silhouettes as *binary* masks that provide supervision on the projected occupancy, as adopted in most previous 3D-unsupervised reconstruction methods [18, 25, 26, 34]. This class of methods aims to optimize for the 3D shapes such that the projected occupancy maps are maximally matched across viewpoints. However, 2D binary occupancy maps offer little geometric supervision of the 3D shape surfaces except at pixel locations where the occupancy map changes value.

Our key insight is that the geometric interpretation of 2D silhouettes has a potentially richer explicit connection to the 3D shape surface $\mathcal{S}$, since 2D silhouettes result from the direct projection of the generating contours on $\mathcal{S}$. We can thus take advantage of the *2D distance transform* on the silhouettes, where each pixel encodes the minimum distance to the 2D silhouette(s) instead of binary occupancy. 2D distance transform is a deterministic operation on binary masks [5], and thus the output contains the same amount of information; however, such information about the geometry is dispersed to all pixels of an image. This allows us to treat 2D distance transform maps as "projections" of 3D signed distance functions (SDF), providing richer supervision on 3D shapes so that all pixels can contribute.

We discuss necessary conditions on the validity of 3D SDFs constrained by the given 2D distance transform maps, where we focus on the set of pixels exterior to the silhouette (denoted as $\mathcal{X}$). We denote $\mathbf{u} \in \mathbb{R}^2$ as the pixel coordinates and $z \in \mathbb{R}$ as projective depth. Furthermore, we denote $\mathcal{D} : \mathbb{R}^2 \to \mathbb{R}$ as the *(Euclidean) distance transform*, where $\mathcal{D}(\mathbf{u})$ encodes the minimum distance of pixel coordinates $\mathbf{u}$ to the 2D silhouette. We assume the camera principal point is at $\mathbf{0} \in \mathbb{R}^2$.

**Proposition.** *If $f$ is a valid 3D SDF, then under (calibrated) perspective cameras,*

$$f(z\bar{\mathbf{u}}) \geq b(z; \mathbf{u}) = z \cdot \left\| \bar{\mathbf{u}} - \frac{\bar{\mathbf{v}}^\top \bar{\mathbf{u}}}{\bar{\mathbf{v}}^\top \bar{\mathbf{v}}} \bar{\mathbf{v}} \right\|_2 \quad \forall z \geq 1, \ \mathbf{u} \in \mathcal{X}, \tag{1}$$

*where*

$$\mathbf{v} = \left( 1 + \frac{\mathcal{D}(\mathbf{u})}{\|\mathbf{u}\|_2} \right) \mathbf{u} \tag{2}$$

*is the 2D point on the $\mathbf{u}$-centered circle of radius $\mathcal{D}(\mathbf{u})$ while being farthest away from the principle point, $\bar{\mathbf{u}} = [\mathbf{u}; 1] \in \mathbb{R}^3$ and $\bar{\mathbf{v}} = [\mathbf{v}; 1] \in \mathbb{R}^3$ are the respective homogeneous coordinates of $\mathbf{u}$ and $\mathbf{v}$, and $b(z; \mathbf{u})$ is a lower bound on the SDF value at 3D point location $z\bar{\mathbf{u}}$.*

*Proof outline.* By the definition of distance transforms, any 2D point within the circle of radius $\mathcal{D}(\mathbf{u})$ centered at $\mathbf{u}$ must be free space. The back-projection of this circle from the camera center forms a

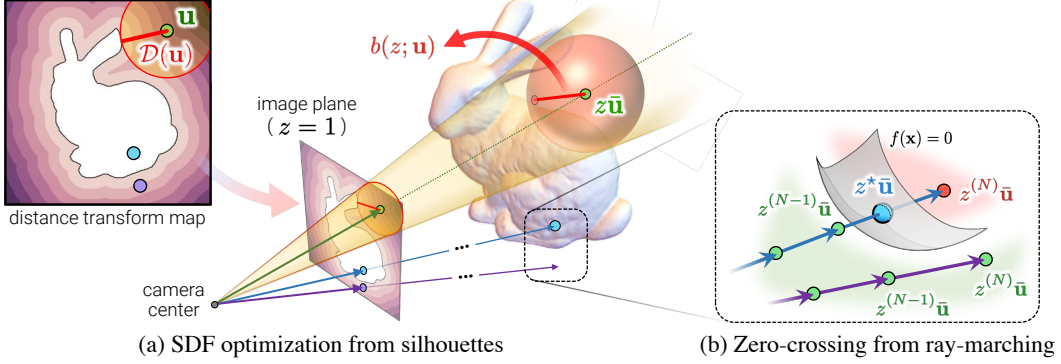

(a) SDF optimization from silhouettes

(b) Zero-crossing from ray-marching

Figure 2: **Learning 3D SDFs from 2D images.** (a) For each pixel $\mathbf{u}$ exterior to the 2D silhouette and its distance transform value $\mathcal{D}(\mathbf{u})$, any 2D point inside the red circle (of radius $\mathcal{D}(\mathbf{u})$) must also be exterior to the silhouette. The back-projection of this circle, forming a cone in the 3D space, must correspondingly be free space. When $\mathbf{u}$ is back-projected into the 3D space to depth $z$, a lower bound $b(z; \mathbf{u})$ on the SDF value can be computed as the radius of the sphere centered at $z\bar{\mathbf{u}}$ and inscribed by the cone (see the supplementary material for detailed derivations). (b) For each pixel inside the 2D silhouette, the closest zero-crossing $z^\star\bar{\mathbf{u}}$ with the surface $\mathcal{S} = \{\mathbf{x} \in \mathbb{R}^3 \mid f(\mathbf{x}) = 0\}$ can be determined via the bisection method between the last two traced points $z^{(N-1)}\bar{\mathbf{u}}$ and $z^{(N)}\bar{\mathbf{u}}$, where opposite signs are encouraged via loss functions in (7) (indicated by the colors on the points).

cone (oblique if $\mathbf{u} \neq \mathbf{0}$), where any 3D point within must also be free space. Therefore, the radius of the sphere centered at $z\bar{\mathbf{u}}$ and inscribed by the cone serves as a lower bound $b(z; \mathbf{u}) = \|z\bar{\mathbf{u}} - z'\bar{\mathbf{v}}\|_2$ for $f(z\bar{\mathbf{u}})$, where $z'\bar{\mathbf{v}}$ is the tangent point of the sphere and the conical surface (with $\mathbf{v}$ on the 2D circle back-projected to depth $z'$). One can thus find $b(z; \mathbf{u})$ by solving the minimization problem

$$\min_{\mathbf{v}, z' \geq 0} \|z\bar{\mathbf{u}} - z'\bar{\mathbf{v}}\|_2^2 \quad \text{subject to} \quad \|\mathbf{v} - \mathbf{u}\|_2 = \mathcal{D}(\mathbf{u}) \ . \tag{3}$$

The first-order optimality conditions on $\mathbf{v}$ and $z'$ in (3) leads to the closed-form solution in (1). $\quad\square$

While we leave the full mathematical proof to the supplementary material, a visual sketch of the above proposition is shown in Fig. 2(a). This states that for any 3D point $z\bar{\mathbf{u}} \in \mathbb{R}^3$, we can always find a lower bound $b(z; \mathbf{u})$ on its possible SDF value from the given distance transform map. It can also be shown that $b(z; \mathbf{u}) = \mathcal{D}(\mathbf{u})$ for the special case of orthographic camera models.

We take advantage of (1) to optimize $f$, which we parametrize with $\boldsymbol{\theta}$. At each training iteration, we randomly sample $M$ depth values $\tilde{z}$ for each pixel $\mathbf{u}$ and formulate the loss function as

$$\mathcal{L}_{\text{SDF}}(\boldsymbol{\theta}) = \frac{\sum_{\mathbf{u} \in \mathcal{X}} w(\mathbf{u}) \cdot \sum_{\tilde{z}} \max\left(0, b(\tilde{z}; \mathbf{u}) - f_{\boldsymbol{\theta}}(\tilde{z}\bar{\mathbf{u}})\right)}{\sum_{\mathbf{u} \in \mathcal{X}} w(\mathbf{u})} \ . \tag{4}$$

We focus the loss more near the silhouettes with the weighting function $w(\mathbf{u}) = \frac{1}{\mathcal{D}(\mathbf{u})}$, decaying with the distance transform value. This is similar to the importance sampling strategy from recent works [26, 20] to improve fidelity of the 3D reconstruction or instance segmentation results.

## 3.2 Rendering Implicit 3D Surfaces

To optimize the 3D implicit surfaces from image data, a differentiable rendering function is required to interpret the continuous 3D feature space. Rendering shape surfaces involves solving for the pixel-wise depth $z$ that corresponds to the 3D point stemming the appearance at pixel coordinates $\mathbf{u}$. Assuming perspective camera models, the problem can be mathematically defined as

$$z^\star = \arg\min_{z \geq 1} z \quad \text{subject to} \quad z\bar{\mathbf{u}} \in \mathcal{S} \ . \tag{5}$$

In other words, rendering $\mathcal{S}$ requires solving for the zero-crossing of each ray of sight closest to the camera center. Ray-casting approaches (*e.g.* sphere tracing [11] and volume rendering [22]) are classical rendering techniques for implicit surfaces in computer graphics, which have also inspired recent differentiable rendering methods [39, 27, 34, 32] for training neural networks with image data.

We build our rendering framework upon the differentiable ray-marching algorithm introduced in Scene Representation Networks (SRN) [39]. At the high level, SRN finds multi-view correspondences implicitly by searching for the terminating 3D point that would result in the most consistent RGB prediction across different viewpoints. SRN ray-marches each pixel from the camera center to predict the 3D geometry through a finite series of learnable steps. Starting at initial depth $z^{(0)}$ for pixel coordinates $\mathbf{u}$, the $j$-th ray-marching step and the update rule can be compactly written as

$$\Delta z^{(j)} = \left| h_{\boldsymbol{\psi}}\big(z^{(j)}\bar{\mathbf{u}}; \boldsymbol{\eta}^{(j)}\big)\right| , \tag{6}$$
$$z^{(j+1)} \leftarrow z^{(j)} + \Delta z^{(j)} , \qquad j \in \{0 \ldots N-1\} ,$$

where $h_{\boldsymbol{\psi}} : \mathbb{R}^3 \to \mathbb{R}$ (parametrized by $\boldsymbol{\psi}$) consists of an MLP and an LSTM cell [13], and $\boldsymbol{\eta}^{(j)}$ summarizes the LSTM states for the $j$-th step, updated internally within in the LSTM cell. We take the absolute value on the output to ensure the ray marches away from the camera. The iterative update on the depth $z$ is repeated $N$ times until the final 3D point $z^{(N)}\bar{\mathbf{u}}$ is reached. Differentiable ray-marching can be viewed as a form of learned gradient descent [1] solving for the problem in (5), which has seen recent success in many applications involving bilevel optimization problems [7, 40, 30].

The ray-marched depth predictions from SRN, however, are view-dependent without guarantee that the resulting 3D geometry would be truly consistent across viewpoints. This can be problematic especially for untextured regions in the images, potentially leading to ambiguous multi-view correspondences being associated. By explicitly introducing the shape surface $\mathcal{S}$, we can resolve for such ambiguity and constrain the final depth prediction $z^{\star}$ to fall on $\mathcal{S}$. Therefore, we employ a second-stage bilevel optimization procedure following the original differentiable ray-marching algorithm. We constrain $z^{\star}$ to fall within the last two ray-marching steps by encouraging negativity on the last ($N$-th) step (modeling a shape interior point) and positivity on the rest (modeling free space along the ray). If the conditions $f(z^{(N)}\bar{\mathbf{u}}) < 0$ and $f(z^{(N-1)}\bar{\mathbf{u}}) > 0$ are satisfied, there must exist $z^{(N-1)} < z^{\star} < z^{(N)}$ such that $f(z^{\star}\bar{\mathbf{u}}) = 0$ from the continuity of the implicit function $f$. An illustration of the above concept is provided in Fig. 2(b). We impose the penalty with margin $\varepsilon$ as

$$\mathcal{L}_{\text{ray}}(\boldsymbol{\theta}, \boldsymbol{\psi}) = \sum_{\mathbf{u}} \sum_{j=0}^{N} \max\Big(0, \alpha(\mathbf{u}) \cdot f_{\boldsymbol{\theta}}\big(z_{\boldsymbol{\psi}}^{(j)}(\mathbf{u})\bar{\mathbf{u}}\big) + \varepsilon\Big), \quad \alpha(\mathbf{u}) = \begin{cases} 1 & \text{if } \mathbf{u} \in \mathcal{X}^{\mathsf{c}} \text{ and } j = N \\ -1 & \text{otherwise} \end{cases} \tag{7}$$

by noting that the ray-marched depth values $z^{(j)}$ are dependent on $h$ and thus parametrized by $\boldsymbol{\psi}$. We denote $\mathcal{X}^{\mathsf{c}}$ as the complement set of $\mathcal{X}$, corresponding to the set of pixels inside the silhouettes. In practice, we also apply importance weighting using the same strategy described in Sec. 3.1. Finally, we solve for $z_{\boldsymbol{\theta}, \boldsymbol{\psi}}^{\star}$ using the bisection method on $f_{\boldsymbol{\theta}}$, whose unrolled form is trivially differentiable.

The rendered pixel at the image coordinates $\mathbf{u}$ can subsequently be expressed as $\widehat{\mathcal{I}}(\mathbf{u}) = g_{\boldsymbol{\phi}}(z_{\boldsymbol{\theta}, \boldsymbol{\psi}}^{\star}\bar{\mathbf{u}})$, where $g_{\boldsymbol{\phi}} : \mathbb{R}^3 \to \mathbb{R}^3$ (parametrized by $\boldsymbol{\phi}$) predicts the RGB values at the given 3D location. We optimize the rendered RGB appearance against the given image $\mathcal{I}$ with the rendering loss

$$\mathcal{L}_{\text{RGB}}(\boldsymbol{\theta}, \boldsymbol{\phi}, \boldsymbol{\psi}) = \sum_{\mathbf{u}} \left\|\widehat{\mathcal{I}}(\mathbf{u}) - \mathcal{I}(\mathbf{u})\right\|_2^2 = \sum_{\mathbf{u}} \left\|g_{\boldsymbol{\phi}}\big(z_{\boldsymbol{\theta}, \boldsymbol{\psi}}^{\star}(\mathbf{u})\bar{\mathbf{u}}\big) - \mathcal{I}(\mathbf{u})\right\|_2^2 , \tag{8}$$

where we write the depth $z_{\boldsymbol{\theta}, \boldsymbol{\psi}}^{\star}(\mathbf{u})$ as a function of the pixel coordinates $\mathbf{u}$. Ideally, $z^{\star}$ should be dependent only on $\boldsymbol{\theta}$, which parametrizes the surface $\mathcal{S}_{\boldsymbol{\theta}} = \{\mathbf{x} \in \mathbb{R}^3 \mid f_{\boldsymbol{\theta}}(\mathbf{x}) = 0\}$; however, the extra dependency on $\boldsymbol{\psi}$ allows the ray-marching procedure to be much more computationally efficient as the step sizes are learned. This also poses an advantage over recent approaches based on classical sphere tracing [27] or volume rendering [34, 32], whose precision of differentiable rendering directly depends on the number of evaluations called on the implicit functions, making them inefficient.

Although SRN was originally designed to learn novel view synthesis from multi-view images, we found it to learn also from single images (in a category-specific setting). We believe this is because correspondences across individual objects still exist at the *semantic* level in single-view training, albeit unavailable at the pixel level. We take advantage of this observation for SDF-SRN. In our case, the hypernetwork learns implicit features that best explains the object semantics within the category; in turn, the ray-marching process discovers and associates implicit semantic correspondences in 3D, such that the ray-marched surfaces are semantically interpretable across all images. Therefore, shape/depth ambiguities can be resolved by learning to recover the appearance (with $\mathcal{L}_{\text{RGB}}$ here), a classical but important cue for disambiguating 3D geometry. This allows SDF-SRN to learn a strong category-specific object prior, making it trainable even from single-view images. Recent works have also shown initial success in this regard with 3D meshes [15] and 3D keypoints [42].

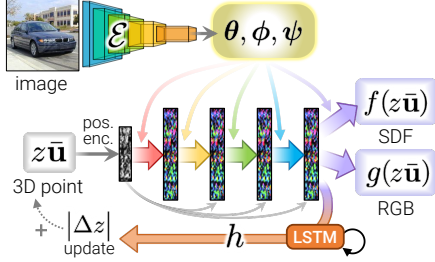

Figure 3: The implicit functions $f$, $g$, and $h$ in SDF-SRN share an MLP backbone with their parameters encoded by $\mathcal{E}$.

Table 1: **Quantitative results** on multi-view ShapeNet data *without* viewpoint association from CAD model correspondences. Note that SDF-SRN even outperforms DVR supervised with depth from visual hull [34]. All numbers are scaled by 10 (the lower the better).

| Category | airplane | | car | | chair | |
|---|---|---|---|---|---|---|
| | accur. | cover. | accur. | cover. | accur. | cover. |
| SoftRas [25] | 0.250 | 0.222 | 0.356 | 0.302 | 0.690 | 0.491 |
| DVR [34] | 1.795 | 0.258 | 1.538 | 0.432 | 1.274 | 0.699 |
| SDF-SRN (ours) | **0.193** | **0.154** | **0.141** | **0.144** | **0.352** | **0.315** |
| DVR (w/ depth) | 0.320 | 0.171 | 0.184 | 0.181 | 0.322 | 0.330 |

### 3.3 Implementation

We use an image encoder $\mathcal{E}$ as a hypernetwork [10] to predict $\boldsymbol{\theta}$, $\boldsymbol{\phi}$, and $\boldsymbol{\psi}$ (the respective function parameters of $f$, $g$, and $h$), written as $(\boldsymbol{\theta}, \boldsymbol{\phi}, \boldsymbol{\psi}) = \mathcal{E}(\mathcal{I}; \boldsymbol{\Phi})$ where $\boldsymbol{\Phi}$ is the neural network weights. The implicit 3D shape $\mathcal{S}_{\boldsymbol{\theta}}$ thus corresponds to the reconstructed 3D shape from image $\mathcal{I}$, which is learned in an object-centric coordinate system. During optimization, $\mathcal{S}_{\boldsymbol{\theta}}$ is transformed to the camera frame at camera pose $(\mathbf{R}, \mathbf{t})$ corresponding to the given image $\mathcal{I}$, rewritten in a parametrized form as

$$\mathcal{S}'_{\boldsymbol{\theta}} = \mathcal{S}_{\boldsymbol{\theta}}(\mathbf{R}, \mathbf{t}) = \{\mathbf{x} \in \mathbb{R}^3 \mid f_{\boldsymbol{\theta}}(\mathbf{R}\mathbf{x} + \mathbf{t}) = 0\} . \tag{9}$$

A special property of SDFs is their differentiability with a gradient of unit norm, satisfying the eikonal equation $\|\nabla f\|_2 = 1$ (almost everywhere) [36]. Therefore, we also encourage our learned implicit 3D representation to satisfy the eikonal property by imposing the penalty

$$\mathcal{L}_{\text{eik}}(\boldsymbol{\theta}) = \sum_{\widetilde{\mathbf{x}}} \left\| \|\nabla f_{\boldsymbol{\theta}}(\widetilde{\mathbf{x}})\|_2 - 1 \right\|_2^2 , \tag{10}$$

where $\widetilde{\mathbf{x}} \in \mathbb{R}^3$ is uniformly sampled from the 3D region of interest. Recent works on learning SDF representations have also sought to incorporate similar regularizations on implicit shapes [9, 14].

To summarize, given a dataset of $D$ tuples $\{(\mathcal{I}, \mathcal{X}, \mathbf{R}, \mathbf{t})_d\}_{d=1}^D$ that consists of the RGB images, 2D silhouettes and camera poses, we train the network $\mathcal{E}$ end-to-end by optimizing the overall objective

$$\mathcal{L}_{\text{all}}(\boldsymbol{\theta}, \boldsymbol{\phi}, \boldsymbol{\psi}) = \lambda_{\text{SDF}}\mathcal{L}_{\text{SDF}}(\boldsymbol{\theta}) + \lambda_{\text{RGB}}\mathcal{L}_{\text{RGB}}(\boldsymbol{\theta}, \boldsymbol{\psi}) + \lambda_{\text{ray}}\mathcal{L}_{\text{ray}}(\boldsymbol{\theta}, \boldsymbol{\phi}, \boldsymbol{\psi}) + \lambda_{\text{eik}}\mathcal{L}_{\text{eik}}(\boldsymbol{\theta}) \tag{11}$$

by noting that $\boldsymbol{\theta}$, $\boldsymbol{\phi}$, and $\boldsymbol{\psi}$ are predicted by the hypernetwork $\mathcal{E}(\mathcal{I}; \boldsymbol{\Phi})$.

## 4 Experiments

**Architectural details.** We use ResNet-18 [12] followed by fully-connected layers as the encoder. We implement the implicit functions $f_{\boldsymbol{\theta}}$, $g_{\boldsymbol{\phi}}$ and $h_{\boldsymbol{\psi}}$ as a shared MLP backbone (Fig. 3), with $f$ and $g$ connecting to shallow heads and $h$ taking the backbone feature as the LSTM input (we do not predict the LSTM parameters with $\mathcal{E}$). The MLP backbone takes a 3D point with positional encoding [32] as input (also added as intermediate features to the hidden layers), which we find to help improve the reconstruction quality. We leave other architectural details to the supplementary material.

**Training settings.** For a fair comparison, we train all networks with the Adam optimizer [19] with a learning rate of $10^{-4}$ and batch size 16. We choose $M = 5$ points for $\mathcal{L}_{\text{SDF}}$ and set the margin $\varepsilon = 0.01$ when training SDF-SRN. Unless otherwise specified, we choose the loss weights to be $\lambda_{\text{RGB}} = 1$, $\lambda_{\text{SDF}} = 3$, $\lambda_{\text{eik}} = 0.01$; we set $\lambda_{\text{ray}}$ to be 1 for the last marched point and 0.1 otherwise. For each training iteration, we randomly sample 1024 pixels $\mathbf{u}$ from each image for faster training.

**Evaluation criteria.** To convert implicit 3D surfaces $\mathcal{S}$ to explicit 3D representations, we sample SDF values at a voxel grid of resolution $128^3$ and extract the 0-isosurface with Marching Cubes [29] to obtain watertight 3D meshes. We evaluate by comparing uniformly sampled 3D points from the mesh predictions to the ground-truth point clouds with the bidirectional metric of Chamfer distance, measuring different aspects of quality: shape accuracy and surface coverage [31, 24].

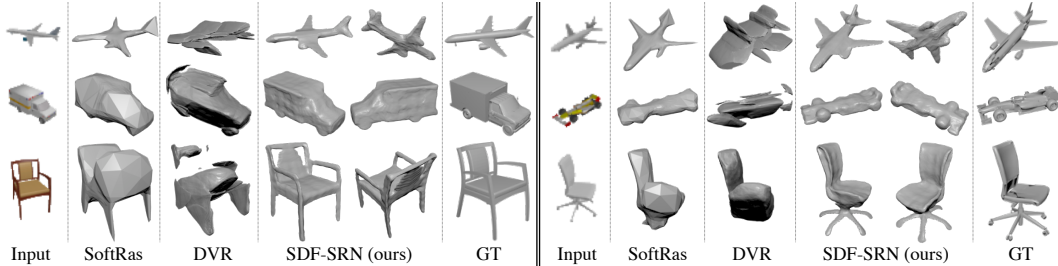

Input&emsp;SoftRas&emsp;DVR&emsp;SDF-SRN (ours)&emsp;GT&emsp;&emsp;Input&emsp;SoftRas&emsp;DVR&emsp;SDF-SRN (ours)&emsp;GT

Figure 4: **Qualitative results** of ShapeNet 3D reconstruction from *single-view training* on multi-view data (*i.e.* no multi-view association is available). Both SoftRas and DVR learns to fit to the silhouettes but fail to learn reasonable 3D reconstruction from independent images, showing their reliance on multi-view constraints. SDF-SRN, in contrast, does not suffer from such limitation and reconstructs 3D objects with high fidelity and successfully recovers the target shape topologies.

Table 2: **Performance analysis** of ShapeNet chairs on the amount of CAD models available as training data under *single*-view supervision. All numbers are scaled by 10 (the lower the better).

| # CADs | SoftRas [25] | | DVR [34] | | Ours | |
|---|---|---|---|---|---|---|
| | accur. | cover. | accur. | cover. | accur. | cover. |
| 500 | 0.550 | 0.508 | 1.298 | 0.674 | **0.475** | **0.422** |
| 1K | 0.547 | 0.551 | 1.284 | 0.701 | **0.442** | **0.385** |
| 2K | 0.522 | 0.481 | 1.268 | 0.609 | **0.423** | **0.349** |
| 4.7K (all) | 0.510 | 0.471 | 1.367 | 1.135 | **0.401** | **0.329** |

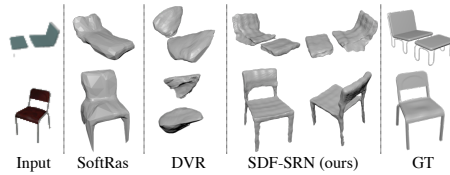

Input&emsp;SoftRas&emsp;DVR&emsp;SDF-SRN (ours)&emsp;GT

Figure 5: SDF-SRN recovers reasonable 3D shapes and topologies even from *single-view* supervision, where each object instance appears only once in the training set.

## 4.1 ShapeNet Objects

**Datasets.** We evaluate our method on the airplane, car, and chair categories from ShapeNet v2 [3], which consists of 4045, 3533, and 6778 CAD models respectively. We benchmark with renderings provided by Kato *et al*. [18], where the 3D CAD models were rendered at 24 uniform viewpoints to $64 \times 64$ images. We split the dataset into training/validation/test sets following Yan *et al*. [46].

**Experimental settings.** We consider a category-specific setting and compare against two baseline methods for learning-based 3D reconstruction without 3D supervision: (a) Differentiable Volumetric Rendering (DVR) [34], a state-of-the-art method for learning implicit occupancy functions, and (b) Soft Rasterizer (SoftRas) [25], a state-of-the-art method for learning 3D mesh reconstruction. We assume known camera poses as with previous works. We also set $\lambda_{\text{SDF}} = 1$ for the airplane category.

Previous works on learning 3D reconstruction from multi-view 2D supervision, including SoftRas and DVR, have assumed additional known *CAD model correspondences*, *i.e.* one knows a priori which images correspond to the same CAD model. This allows a training strategy of pairing images with viewpoints selected from the same 3D shape, making the problem more constrained and much easier to optimize. Practically, however, one rarely encounters the situation where such viewpoint associations are explicitly provided while also having silhouettes annotated. In pursuit of an even more practical scenario of 3D-unsupervised learning, we train all models where all rendered images are treated *independently*, *without* the knowledge of multi-view association. This is equivalent to *single-view training* on multi-view data, which is more challenging than "multi-view supervision", and can also be regarded as an autoencoder setting with additional camera pose supervision.

**Results.** We evaluate on the multi-view rendered images and train all airplane and car models for 100K iterations and all chair models for 200K iterations. The results are reported in Table 1 and visualized in Fig. 4. SDF-SRN outperforms both baseline methods by a wide margin and recovers accurate 3D object shapes, despite no explicit information of multi-view association being available. On the other hand, SoftRas and DVR trained on individual images without such multi-view constraint do not learn shape regularities within the category and cannot resolve for shape ambiguities, indicating their high reliance on viewpoint association of the same CAD model. Furthermore, SDF-SRN even outperforms DVR additionally supervised with depth information extracted from visual hulls [34].

We further analyze the performances under the more practical *single-view* supervision setting on *single-view* data, where only one image (at a random viewpoint) per CAD model is included in the training set. We choose the chair category and train each model for 50K iterations; since there are much fewer available training images, we add data augmentation with random color jittering on the fly to reduce overfitting. Table 2 shows that SDF-SRN also outperforms current 3D-unsupervised methods in this challenging setting even trained with scarce data (500 images), which further improves as more training data becomes available. In addition, SDF-SRN is able to recover accurate shape topologies (Fig. 5) even under this scenario, giving hints to its applicability to real-world images.

Table 3: Ablation studies of SDF-SRN (conducted on ShapeNet chairs). We additionally evaluate the performance with test-time optimization on the latent code over the fully trained networks if the camera poses were known a priori.

| | accur. | cover. |
|---|---|---|
| binary occupancy | 0.425 | 0.790 |
| w/o rendering loss $\mathcal{L}_{\text{RGB}}$ | 0.353 | 0.554 |
| w/o importance weighting | 0.483 | 1.168 |
| w/o positional encoding | 0.444 | 0.351 |
| full model (SDF-SRN) | **0.352** | **0.315** |
| w/ test-time optimization | **0.332** | **0.303** |

**Ablative analysis.** We discuss the effects of different components essential to SDF-SRN (under the multi-view data setup) in Table 3. Training with binary cross-entropy (classifying 3D occupancy) results in a significant drop of performance, which we attribute to the nonlinear nature of MLPs using discontinuous binary functions as the objective. Table 3 also shows the necessity of differentiable rendering ($\mathcal{L}_{\text{RGB}}$) for resolving shape ambiguities (*e.g.* concavity cannot be inferred solely from silhouettes), importance weighting for learning finer shapes that aligns more accurately to silhouettes, and positional encoding of 3D point locations to extract more high-frequency details from images. We additionally show that if the camera pose were given, one could optimize the same losses in (11) for the latent code over the trained network, further reducing prediction uncertainties and improving test-time reconstruction performance [37, 24, 39].

## 4.2 Natural Images

**Dataset.** We demonstrate the efficacy of our method on PASCAL3D+ [45], a 3D reconstruction benchmarking dataset of real-world images with ground-truth CAD model annotations. We evaluate on the airplane, car, and chair categories from the ImageNet [6] subset, which consists of 1965, 5624, and 1053 images respectively. PASCAL3D+ is challenging in at least 3 aspects: (a) the images are much scarcer than ShapeNet renderings since human annotations of 3D CAD models is laborious and difficult to scale up; (b) object instances appear only once without available multi-view observations to associate with; (c) the large variations of textures and lighting in images makes it more difficult to learn from. We assume weak-perspective camera models and use the provided 2D bounding boxes to normalize the objects, square-crop the images, and resize them to $64 \times 64$.

**Experimental settings.** We compare against DVR [34] as well as Category-specific Mesh Reconstruction (CMR) [15], which learns 3D mesh reconstruction from static images. Only RGB images and object silhouettes are available during training; we do not assume any available pretrained 2.5D/3D shape priors from external 3D datasets [41, 44]. We initialize the meshes in CMR with a unit sphere; for a fair comparison, we consider CMR using ground-truth camera poses instead of predicting them from keypoints, which was also reported to yield better performance. We train each model for 30K iterations, augmenting the dataset with random color jittering and image rescaling (uniformly between $[0.8, 1.2]$). We set $\lambda_{\text{RGB}} = 10$ and $\lambda_{\text{eik}} = 1$ for SDF-SRN. We evaluate quantitatively with shape predictions registered to the ground truth using the Iterative Closest Point algorithm [2].

**Results.** We present qualitative results in Fig. 6. Even though the texture and lighting variations in PASCAL3D+ makes it more difficult than ShapeNet to associate correspondences across images, SDF-SRN is able to recover more accurate shapes and topologies from the images compared to the baseline methods. CMR has difficulty learning non-convex shape parts (*e.g.* plane wings), while DVR has difficulty learning meaningful shape semantics within category. We note that SDF-SRN also suffers slightly from shape ambiguity (*e.g.* sides of cars tend to be concave); nonetheless, SDF-SRN still outperforms CMR and DVR quantitatively by a wide margin (Table 4). We also visualize colors and surface normals from the reconstructions rendered at novel viewpoints (Fig. 7), showing that SDF-SRN is able to capture meaningful semantics from singe images such as object symmetry.

Finally, we note that recent research has shown possible to obtain camera poses from more practical supervision sources (*e.g.* 2D keypoints [35, 43]), which could allow learning shape reconstruction from larger datasets that are much easier to annotate. We leave this to future work.

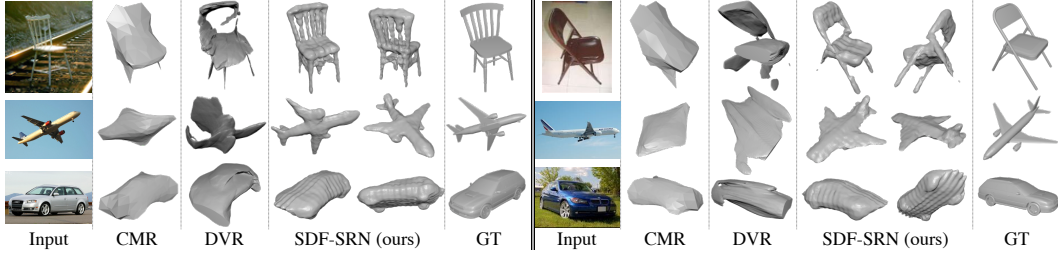

| Input | CMR | DVR | SDF-SRN (ours) | GT | Input | CMR | DVR | SDF-SRN (ours) | GT |

Figure 6: **Qualitative results** from PASCAL3D+ reconstruction. Compared to the two baseline methods, SDF-SRN recovers significantly more accurate 3D shapes and topologies from the images. Both CMR and DVR struggle to associate meaningful shape regularities within category, while CMR additionally suffers from topological limitations due to its mesh-based nature, as with SoftRas.

Table 4: **Quantitative comparison** on PASCAL3D+. All models were trained from scratch solely on the images and silhouettes, without utilizing 2.5D/3D shape priors pretrained from external 3D datasets. SDF-SRN consistently outperforms both baseline methods. All numbers are scaled by 10 (the lower the better).

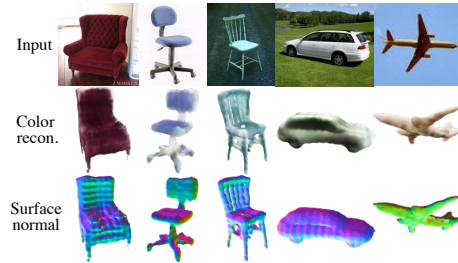

Figure 7: PASCAL3D+ reconstruction with color and surface normal predictions.

| Category | airplane | | car | | chair | |
|---|---|---|---|---|---|---|
| | accur. | cover. | accur. | cover. | accur. | cover. |
| CMR [15] | 0.625 | 0.803 | 0.474 | 0.623 | 1.396 | 1.168 |
| DVR [34] | 1.483 | 0.916 | 1.493 | 0.795 | 3.356 | 2.251 |
| Ours | **0.582** | **0.543** | **0.391** | **0.402** | **0.478** | **0.398** |

## 5   Conclusion

We have introduced SDF-SRN for learning dense 3D reconstruction from static image collections. Our framework learns SDF shape representations from distance transformed 2D silhouettes with improved differentiable rendering. SDF-SRN does not rely on associated multi-view supervision and learns from *single*-view images, demonstrating compelling 3D reconstruction results even on challenging natural images. We believe this is an exciting direction and opens up avenues for learning 3D reconstruction from larger real-world datasets with more practical supervision.

**Acknowledgements.** We thank Deva Ramanan, Abhinav Gupta, Andrea Vedaldi, Ioannis Gkioulekas, Wei-Chiu Ma, Ming-Fang Chang, Yufei Ye, Stanislav Panev, Rahul Venkatesh, and the reviewers for helpful discussions and feedback on the paper. CHL is supported by the NVIDIA Graduate Fellowship. This work was supported by the CMU Argo AI Center for Autonomous Vehicle Research.

## Broader Impact

Our proposed framework, SDF-SRN, allows for learning dense 3D geometry of object categories from real-world images using annotations (*i.e.* 2D silhouettes) that can be feasibly obtained at a large scale. Computer vision increasingly needs to perform 3D geometric reasoning from images, such as when an autonomous vehicle encounters a vehicle in the streets. To avoid catastrophe, the car must not only detect the existence of the vehicle but also exactly determine its spatial extent in the 3D world. Similarly, robots and drones are increasingly deployed in unconstrained environments where they must safely manipulate and avoid 3D objects. Health professionals are increasingly using computer vision to interpret 2D scans/imagery in 3D. Breakthroughs in dense geometric reasoning could allow these researchers to extract unprecedented detail from visual data.

This work also offers up exciting new opportunities in the area of computer graphics for 3D content creation, where the laborious process of creating 3D models and animations could be significantly simplified. This could reduce the time and money costs required for many industrial applications (*e.g.* involving virtual reality). We should note that all new technologies have the potential for misuse, and our framework SDF-SRN is no different in this regard. However, we strongly believe the myriad of possible societal and economic benefits of our work vastly outweigh such risks.

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
