[Supplementary Material · supp.pdf]

# SDF-SRN: Learning Signed Distance
# 3D Object Reconstruction from Static Images
## Supplementary Document

**Chen-Hsuan Lin**    **Chaoyang Wang**    **Simon Lucey**
Carnegie Mellon University
chlin@cmu.edu    chaoyanw@andrew.cmu.edu    slucey@cs.cmu.edu
https://chenhsuanlin.bitbucket.io/signed-distance-SRN/

## A  Derivation of the Proposition

### A.1  Formal Proof

We provide more detailed derivations of the lower bound $b(z; \mathbf{u})$. For clarity, we briefly restate the proof outline from the main paper first before going into the main proof.

For pixel coordinates $\mathbf{u}$ on the image plane with distance transform value $\mathcal{D}(\mathbf{u})$, the set of 2D points $\{\mathbf{v} \mid \|\mathbf{v} - \mathbf{u}\|_2 \le \mathcal{D}(\mathbf{u})\}$ within the circle of radius $\mathcal{D}(\mathbf{u})$ centered at $\mathbf{u}$ must be exterior to the 2D silhouette. It immediately follows that the set of 3D points $\{z'\bar{\mathbf{v}} \mid z' \ge 0, \|\mathbf{v} - \mathbf{u}\|_2 \le \mathcal{D}(\mathbf{u})\}$ within the (oblique) cone formed by back-projecting the circle from the camera center must be free space. For a 3D point $z\bar{\mathbf{u}}$, the SDF value $f(z\bar{\mathbf{u}})$ is thus lower-bounded by the radius of the sphere centered at $z\bar{\mathbf{u}}$ while being inscribed by the cone. Let $b(z; \mathbf{u}) = \|z\bar{\mathbf{u}} - z'\bar{\mathbf{v}}\|_2$ be the lower bound for $f(z\bar{\mathbf{u}})$, where $z'\bar{\mathbf{v}}$ is the tangent point of the sphere and the conical surface (with $\mathbf{v}$ on the 2D circle back-projected to depth $z'$). We aim to find $b(z; \mathbf{u})$ by solving the minimization problem

$$\min_{\mathbf{v}, z' \ge 0} \|z\bar{\mathbf{u}} - z'\bar{\mathbf{v}}\|_2^2 \quad \text{subject to } \|\mathbf{v} - \mathbf{u}\|_2 = \mathcal{D}(\mathbf{u}) \; . \tag{1}$$

First, by noting $\mathbf{u} = (u_x, u_y)$, we reparametrize $\mathbf{v}$ (the set of 2D points on the circle) with $\theta$ to be

$$\mathbf{v} = \begin{bmatrix} u_x + \mathcal{D}(\mathbf{u}) \cos\theta \\ u_y + \mathcal{D}(\mathbf{u}) \sin\theta \end{bmatrix} \; . \tag{2}$$

The problem in (1) thus becomes

$$\min_{\theta, z' \ge 0} \left\| z \begin{bmatrix} u_x \\ u_y \\ 1 \end{bmatrix} - z' \begin{bmatrix} u_x + \mathcal{D}(\mathbf{u}) \cos\theta \\ u_y + \mathcal{D}(\mathbf{u}) \sin\theta \\ 1 \end{bmatrix} \right\|_2^2 \; . \tag{3}$$

Without loss of generality, the first-order optimality condition on $\theta$ is

$$
\begin{aligned}
0 &= 2 \cdot \left( z \begin{bmatrix} u_x \\ u_y \\ 1 \end{bmatrix} - z' \begin{bmatrix} u_x + \mathcal{D}(\mathbf{u}) \cos\theta \\ u_y + \mathcal{D}(\mathbf{u}) \sin\theta \\ 1 \end{bmatrix} \right)^\top \left( -z' \begin{bmatrix} -\mathcal{D}(\mathbf{u}) \sin\theta \\ \mathcal{D}(\mathbf{u}) \cos\theta \\ 0 \end{bmatrix} \right) \\
&= \left( z \begin{bmatrix} u_x \\ u_y \end{bmatrix} - z' \begin{bmatrix} u_x + \mathcal{D}(\mathbf{u}) \cos\theta \\ u_y + \mathcal{D}(\mathbf{u}) \sin\theta \end{bmatrix} \right)^\top \begin{bmatrix} -\sin\theta \\ \cos\theta \end{bmatrix} \\
&= \left( z \begin{bmatrix} u_x \\ u_y \end{bmatrix} - z' \begin{bmatrix} u_x \\ u_y \end{bmatrix} \right)^\top \begin{bmatrix} -\sin\theta \\ \cos\theta \end{bmatrix} = \begin{bmatrix} u_x \\ u_y \end{bmatrix}^\top \begin{bmatrix} -\sin\theta \\ \cos\theta \end{bmatrix} \; ,
\end{aligned}
\tag{4}
$$

leading to $\theta = \tan^{-1} \frac{u_y}{u_x}$ and thus

$$\cos\theta = \frac{u_x}{\|\mathbf{u}\|_2} \quad \text{and} \quad \sin\theta = \frac{u_y}{\|\mathbf{u}\|_2} \ . \tag{5}$$

This indicates the optimality of $\theta$ is only dependent on the (normalized) image coordinates $\mathbf{u}$ while being *independent* of both the given depth $z$ and the variable $z'$. Plugging (5) back into (2) leads to

$$\mathbf{v} = \begin{bmatrix} u_x + \mathcal{D}(\mathbf{u})\dfrac{u_x}{\|\mathbf{u}\|_2} \\ u_y + \mathcal{D}(\mathbf{u})\dfrac{u_y}{\|\mathbf{u}\|_2} \end{bmatrix} = \left(1 + \frac{\mathcal{D}(\mathbf{u})}{\|\mathbf{u}\|_2}\right)\mathbf{u} \ . \tag{6}$$

Having solved for $\mathbf{v}$, the problem in (1) simplifies into

$$\min_{z' \geq 0} \|z\bar{\mathbf{u}} - z'\bar{\mathbf{v}}\|_2^2 \ , \tag{7}$$

which is a linear problem with the solution

$$z' = z \cdot \frac{\bar{\mathbf{u}}^\top \bar{\mathbf{v}}}{\bar{\mathbf{v}}^\top \bar{\mathbf{v}}} \ . \tag{8}$$

Note that $z'$ satisfies the non-negativity constraint by nature; one can verify by plugging (6) into (8).

Finally, the lower bound thus becomes

$$b(z; \mathbf{u}) = \|z\bar{\mathbf{u}} - z'\bar{\mathbf{v}}\|_2 = z \cdot \left\| \bar{\mathbf{u}} - \frac{\bar{\mathbf{v}}^\top \bar{\mathbf{u}}}{\bar{\mathbf{v}}^\top \bar{\mathbf{v}}} \bar{\mathbf{v}} \right\|_2 \tag{9}$$

by noting the expression of $\mathbf{v}$ is given in (6).

### A.2   Orthographic cameras

We show that $b(z; \mathbf{u}) = \mathcal{D}(\mathbf{u})$ under orthographic cameras. Intuitively, one can imagine the camera center to be pulled away from the image to negative infinity, in which case the back-projected cone would approach an unbounded cylinder. Correspondingly, the radius of the inscribed sphere would always be $\mathcal{D}(\mathbf{u})$, irrespective of the queried depth $z$.

The back-projected 3D point (denoted as $\mathbf{x}$) from pixel coordinates $\mathbf{u}$ at depth $z$ has a fixed distance $z - 1$ to the image plane. Denoting $f_c$ as the camera focal length and writing the new depth as a function of $f_c$, we can rewrite the queried 3D point as

$$\mathbf{x} = \frac{z - 1 + f_c}{f_c} \begin{bmatrix} \mathbf{u} \\ f_c \end{bmatrix} \ , \tag{10}$$

which becomes $\mathbf{x} = z\bar{\mathbf{u}}$ when $f_c = 1$. Similarly, we can rewrite the tangent point of the cone and the inscribed sphere

$$\mathbf{x}' = \frac{z' - 1 + f_c}{f_c} \begin{bmatrix} \mathbf{v} \\ f_c \end{bmatrix} \ , \tag{11}$$

which becomes $\mathbf{x}' = z'\bar{\mathbf{v}}$ when $f_c = 1$. The lower bound $b(z; \mathbf{u}) = \|\mathbf{x} - \mathbf{x}'\|_2$ thus becomes

$$\begin{aligned} b(z; \mathbf{u}) &= \|\mathbf{x} - \mathbf{x}'\|_2 \\ &= \left\| \frac{z - 1 + f_c}{f_c} \begin{bmatrix} \mathbf{u} \\ f_c \end{bmatrix} - \frac{z' - 1 + f_c}{f_c} \begin{bmatrix} \mathbf{v} \\ f_c \end{bmatrix} \right\|_2 \ . \end{aligned} \tag{12}$$

For orthographic cameras (where $f_c$ approaches infinity), taking the limit of $f_c \to \infty$ on (12) yields

$$\begin{aligned} \lim_{f_c \to \infty} b(z; \mathbf{u}) &= \lim_{f_c \to \infty} \left\| \frac{z - 1 + f_c}{f_c} \begin{bmatrix} \mathbf{u} \\ f_c \end{bmatrix} - \frac{z' - 1 + f_c}{f_c} \begin{bmatrix} \mathbf{v} \\ f_c \end{bmatrix} \right\|_2 \\ &= \|\mathbf{u} - \mathbf{v}\|_2 = \mathcal{D}(\mathbf{u}) \ . \end{aligned} \tag{13}$$

## B   Dataset

In this section, we provide more details on the datasets used in the experiments.

## B.1 ShapeNet [3]

The dataset split from Yan *et al*. [15] were from ShapeNet v1. As nearly half of the CAD models in the car category were removed in ShapeNet v2, we take the intersection as the final splits for our experiments. The final statistics of ShapeNet CAD models are reported in Table 1. Following standard protocol, we used the validation set for hyperparameter tuning and the test set for evaluation.

The ground-truth point clouds provided by Kato *et al*. [9] were directly sampled from the ShapeNet CAD models. However, many interior details were also included in a significant portion of the CAD models, especially the car category (*e.g.* car seats and steering wheels). Such details cannot be recovered by 3D-unsupervised 3D reconstruction frameworks, including SoftRas [10], DVR [12], and the proposed SDF-SRN, as they are limited to reconstructing the outer (visible) surfaces of objects. Therefore, we use the ground truth provided by Groueix *et al*. [6], whose point clouds were created using a virtual mesh scanner from Wang *et al*. [13]. We re-normalize the point clouds to match that from Kato *et al*. [9], tightly fitting a zero-centered unit cube.

When performing on-the-fly data augmentation in the single-view supervision experiments, we randomly perturb the brightness by $[-20\%, 20\%]$, contrast by $[-20\%, 20\%]$, saturation by $[-20\%, 20\%]$, and hue uniformly in the entire range. We do not perturb the image scales for ShapeNet renderings.

Table 1: Dataset statistics of ShapeNet v2 (number of CAD models).

| Category | train | validation | test | total |
|---|---|---|---|---|
| airplane | 2830 | 809 | 405 | 4044 |
| car | 2465 | 359 | 690 | 3514 |
| chair | 4744 | 678 | 1356 | 6778 |

Table 2: Dataset statistics of PAS-CAL3D+ (number of images).

| Category | train | validation | total |
|---|---|---|---|
| airplane | 991 | 974 | 1965 |
| car | 2847 | 2777 | 5624 |
| chair | 539 | 514 | 1053 |

## B.2 PASCAL3D+ [14]

The PASCAL3D+ dataset is comprised of two subsets from PASCAL VOC [5] and ImageNet [4], labeled with ground-truth CAD models and camera poses. We evaluate on the ImageNet subset as it exhibits much less object occlusions than the PASCAL VOC subset. We note that occlusion handling is still an open problem to all methods (including the proposed SDF-SRN) since appropriate normalization of object scales is required to learn meaningful semantics within category. The statistics of PASCAL3D+ used in the experiments are reported in Table 2.

Since the natural images from PASCAL3D+ come in different image and object sizes, we rescale by utilizing the (tightest) 2D bounding boxes. In particular, we center the object and rescale such that 1.2 times the longer side of the bounding box fits the canonical images (with a resolution of $64 \times 64$); for the car category, we rescale such that the height of the bounding box fits $1/3$ of the canonical image height. When performing on-the-fly data augmentation, we randomly perturb the brightness by $[-20\%, 20\%]$, contrast by $[-20\%, 20\%]$, saturation by $[-20\%, 20\%]$, and hue uniformly in the entire range. We additionally perturb the image scale by $[-20\%, 20\%]$.

We use the ground-truth CAD model and camera pose associated with each image to create the object silhouettes. For evaluation, we create ground-truth point clouds from the 3D CAD models using the virtual mesh scanner from Wang *et al*. [13] (described in Sec. B.1) and rescale by the factor

$$s = \frac{\text{camera focal length}}{\text{camera distance}} \cdot \frac{2}{64} , \qquad (14)$$

which scales the point clouds to match the $[-1, 1]$ canonical image space where the silhouettes lie. Note that the camera parameters provided from the dataset are used only for evaluation. Since there may still exist misalignment mainly due to depth ambiguity, we run the rigid version of the Iterative Closest Point algorithm [2] for 50 iterations to register the prediction to the rescaled ground truth.

## C Architectural and Training Details

We provide a more detailed description of the network architectures of SDF-SRN. As described in the paper, the implicit functions $f_{\boldsymbol{\theta}}$, $g_{\phi}$ and $h_{\psi}$ share an MLP backbone extracting point-wise

deep features for each 3D point. The shared MLP backbone consists of 4 linear layers with 128 hidden units, with layer normalization [1] and ReLU activations in between. The shallow heads of $f$ and $g$ are single linear layers, predicting the SDF and RGB values respectively. The encoder $\mathcal{E}$ is built with a ResNet-18 [7] followed by fully-connected layers, which consists of six 512-unit hidden layers for the ShapeNet [3] experiments and two 256-unit hidden layers for the PASCAL3D+ [14] experiments. We use separate branches of fully-connected layers to predict the weights and biases of each linear layer of the implicit function, *i.e.* the latent code (from the encoder) is passed to 4 sets of fully-connected layers for the 4 linear layers of the MLP backbone, and similarly for the shallow heads. We choose the hidden and output state dimension for the LSTM to be 32 and add a following linear layer to predict the update step for the depth $\delta z$.

Following prior practice [8, 12], we initialize ResNet-18 with weights pretrained with ImageNet [4]. To make the training of SDF-SRN better conditioned, we also pretrain the fully-connected part of the hypernetwork to initially predict an SDF space of a zero-centered sphere with radius $r$. In practice, we randomly sample $\widetilde{\mathbf{z}} \sim \mathcal{N}(\mathbf{0}, \mathbf{I})$ on the hidden latent space to predict the weights $\widetilde{\boldsymbol{\theta}}$ of an implicit function $f_{\widetilde{\boldsymbol{\theta}}}$. Subsequently, we uniformly sample $\widetilde{\mathbf{x}} \in \mathbb{R}^3$ in the 3D space and minimize the loss

$$\mathcal{L}_{\text{pretrain}} = \sum_{\widetilde{\mathbf{x}}} \left\| f_{\widetilde{\boldsymbol{\theta}}}(\widetilde{\mathbf{x}}) - \left( \|\widetilde{\mathbf{x}}\|_2 - r \right) \right\|_2^2 . \tag{15}$$

We choose radius $r = 0.5$ and pretrain the fully-connected layers randomly sampling 10K points for 2000 iterations. We find this pretraining step important for facilitating convergence at the early stage of training and avoiding degenerate solutions.

To encourage high-frequency components to be recovered in the 3D shapes, we take advantage of the positional encoding technique advocated by Mildenhall *et al.* [11]. For each input 3D point $\mathbf{x}$ of the implicit functions $f$, $g$, and $h$, we map $\mathbf{x}$ to higher dimensions with the encoding function

$$\gamma(p) = \left[ p, \cos(2^0 p), \sin(2^0 p), \ldots, \cos(2^{L-1} p), \sin(2^{L-1} p) \right] , \tag{16}$$

where $p$ is applied to all 3D coordinates of $\mathbf{x} = (x, y, z)$ and subsequently concatenated. We choose $L = 6$ in our implementation. More details can be found in the work of Mildenhall *et al.* [11].

## D   Supplementary Video

Please see the attached video `supp.mp4` for a 3D visualization of the 3D reconstruction results on both ShapeNet [3] and PASCAL3D+ [14].

## E   Additional Results

We visualize additional qualitative comparisons for the ShapeNet [3] multi-view supervision experiment in Fig. 1 for airplanes, Fig. 2 for cars, and Fig. 3 for chairs. We again emphasize that these results are from *unknown* multi-view associations, where each image is treated independently during training. SDF-SRN is able to consistently capture meaningful shape semantics within category, while SoftRas [10] and DVR [12] suffer from the lack of viewpoint correspondences. In addition, SDF-SRN can successfully capture various shape topologies that underlies in the images. We also provide additional results of SDF-SRN on natural images (PASCAL3D+ [14]) in Fig. 4 for airplanes, Fig. 5 for cars, and Fig. 6 for chairs.

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

Input&emsp;&emsp;SoftRas&emsp;&emsp;DVR&emsp;&emsp;SDF-SRN (ours)&emsp;&emsp;GT&emsp;&emsp;Input&emsp;&emsp;SoftRas&emsp;&emsp;DVR&emsp;&emsp;SDF-SRN (ours)&emsp;&emsp;GT

Figure 1: Additional results on ShapeNet airplanes.

Input&emsp;&emsp;SoftRas&emsp;&emsp;DVR&emsp;&emsp;SDF-SRN (ours)&emsp;&emsp;GT&emsp;&emsp;Input&emsp;&emsp;SoftRas&emsp;&emsp;DVR&emsp;&emsp;SDF-SRN (ours)&emsp;&emsp;GT

Figure 2: Additional results on ShapeNet cars.

Input&emsp;&emsp;SoftRas&emsp;&emsp;DVR&emsp;&emsp;SDF-SRN (ours)&emsp;&emsp;GT&emsp;&emsp;Input&emsp;&emsp;SoftRas&emsp;&emsp;DVR&emsp;&emsp;SDF-SRN (ours)&emsp;&emsp;GT

Figure 3: Additional results on ShapeNet chairs.

Figure 4: Additional results on PASCAL3D+ airplanes.

Figure 5: Additional results on PASCAL3D+ cars.

Figure 6: Additional results on PASCAL3D+ chairs.