[Reviews · NeurIPS 2020]

Review 1

Summary and Contributions: This paper presents a differentiable rendering framework for implicit surface based on Scene Representation Network [38]. It refines the formuation of [38] by leveraging the distance transformed sihouettes for richer geometric supervision.

Strengths: - An improved differentiable rendering framework for implicit surface is proposed, which can further improve the rendering accuracy of scene representation network [38].

Weaknesses: - The writing of this paper is very misleading. First of all, it claims that it can be trained only using a single viewpoint of the object. In fact, all previous diffrentiable rendering techniques can be trained using a single view of object at training time. However, the reason why multi-view images are used for training in prior works is that single-view images usually lead to ambiguity in the depth direction. The proposed method also suffers from this problem -- it cannot resolve the ambiguity of depth using a single image either. The distrance-transformed silhouette can only provide information on the xy plane - the shape perpendicular to the viewing direction. - I doubt the proposed method can be trained without using any camera information (Line 223, the so called "knowledge of CAD model correspondences"). Without knowing the viewpoint, how is it possible to perform ray marching? How do you know where the ray comes from? - The experiments are not comprehensive and convincing. 1) The comparisons do not seem fair. The performance of DVR is far worse than that in the original DVR paper. Is DVR trained and tested on the same data? What is the code used for evaluation? Is it from the original authors or reimplementation? 2) Though it could be interesting to see how SoftRas performs, it is not very fair to compare SoftRas here as it use a different 3D representation -- mesh. It is well known that mesh representation cannot model arbitrary topology. Thus it is not surprising to see it is outperformed. Since this paper works on implicit surface, it would be more interesting to compare with more state-of-the-art differentiable renderers for implicit surface, i.e. [26],[27],[14], or at least the baseline approach [38]. However, no direct comparisons with these approaches are provided, making it difficult to verify the effectivenss of the proposed approach.

Correctness: Many of the claims are incorrect or overclaimed. Just to name a few. For instance, Line 42-43: "In this paper, we make significant advances on learning dense 3D object reconstruction from single images and silhouettes, without the knowledge of the underlying shape structure or topology." -- This has been already achieved by many of previous works, i.e. [26],[27],[14]. Please refer to the weakness section for more details. Method-wise, the proposed method seems to be able to improve the rendering accuracy. However, the provided experiments are not convincing to verify this point.

Clarity: This work fails to achieve the contributions that claimed in the paper. The entire positioning of this paper is very misleading. Some of the claims are incorrect based on the proposed method. Please refer to the weakness sections for more details.

Relation to Prior Work: This paper is overclaiming its contribution, as elaborated in the weakness section, when discussing its advantages over the previous works. I would highly recommend the writing to be adjusted accordingly in the revisions.

Reproducibility: No

Additional Feedback: It is not clear how the proposed method is able to train network without using any viewpoint information as claimed in the paper. More details are needed to clarify this point. -- post rebuttal -- The authors' rebuttal still cannot convince me that the proposed approach can resolve the ambiguity in the depth direction and thus be able to learn 3D shapes only using a single image for each object during training. This is very important because if the framework cannot resolve the ambiguity in the depth direction, all the contributions and claims of this work do not stand. 1) In the rebuttal, the author claimed that single-view training is possible because they exist at the semantic level and use CMR as a successful example. However, CMR resolves the depth ambiguity using an initial category-specific 3D template, introducing a very strong prior/initial value for the depth. Hence, CMR only needs to learn a plausible deformation of the template without the need to estimate the depth from the ground up. In contrast, the setting of SDF-SRN is totally different in that we have no prior knowledge of the shape but only an image. And thus the success of CMR cannot provide any insight that current framework can be successful. 2) As also pointed out by R4, the 2D SDF loss can only provide supervision on the contour of a one-view visual hull as only the silhouette is used. The ray sampler can only help to sample more efficiently but cannot resolve the ambiguity in the depth direction either. The color loss can be fooled by a flat plane with the exact same texture with the input image. None of these loss terms can resolve the ambiguity of the depth without using multiple views. 3) The authors mentioned in the rebuttal that the ray marching method can learn semantic correspondence, which is very vague. In the presented method, I do not see any part of it that can learn semantic correspondence across objects.


Review 2

Summary and Contributions: The paper introduced a novel formulation to optimize SDF representation from 2D distance transformations, and achieves dense 3D object reconstruction outputs. The paper also proposed a differentiable rendering framework that are suitable for SDF representations. Experimental results on synthetic and real images demonstrate that the proposed approach is significantly better than existing algorithms.

Strengths: The paper is well-written, and contain several professional/beautiful visualizations that can help readers understand the approach. The contributions of combining SRN, SDF prediction, as well as the design of those loss functions are all nicely connected. Experimental results also suggest that the approach achieves great results comparing to existing approaches.

Weaknesses: This is a nicely written paper, but I have some constructive comments, and I hope authors can address those issues in future revisions. - I wonder if the proposed approach will generalize well to objects with extreme poses (such as a chair where the majority of the region is occluded). It seems that under the current formulation, a significant amount of object parts need to be visible. With multi-view supervision, the approach seems to be achieving better performance. How are those viewpoints being chosen in training? - What is the run-time of the proposed approach? I think it may depend on the LSTM steps too. Could authors comment on how those timesteps are determined, and the comment on the output quality improvements along the way? - DVR[33] was not utilizing silhouettes, so I wonder if that is a fair comparison or not. - In real-world applications, the silhouettes are usually not available. Therefore users may need to hand-draw such masks.

Correctness: Yes.

Clarity: Yes, I especially like the diagrams and visualizations of this paper.

Relation to Prior Work: Yes.

Reproducibility: Yes

Additional Feedback:


Review 3

Summary and Contributions: The paper proposes a method for single-view-3d reconstruction. The method is trained with image, silhouette, and camera-pose ground truth. It leverages a distance transform of the silhouettes to train a set of networks for regressing a 3D signed distance function. A hypernetwork predicts from an image weights of MLPs and an LSTM, which in turn predict occupancy and color given 3D points.

Strengths: + The proposed method allows learning without explicit 3D geometry supervision and only from a single view per object. + The method is well described and the paper clearly written. + The method is evaluated against strong baselines and performs better. + Training on just a single view per object as proposed is interesting for learning from novel categories where 3D CAD models for supervision may not be readily available.

Weaknesses: - The dependence on the camera pose somewhat limits the applicability. It also raises the question how much the proposed method relies on high quality camera poses. Similarly, it relies on high quality silhouette images and it is not quite clear how robust the method would be in practical setups with silhouettes coming from e.g. an instance segmentation algorithm. An experiment with different levels of noise for the camera parameters and less than perfect silhouettes would be helpful to asses the robustness in real-world settings. - I would appreciate a more detailed motivation for the hypernetworks. Other work directly optimized the MLPs or conditioned predictions from the MLPs via latent codes. I would suspect those approaches to be more robust than a network predicting network weights. Further, I expect the hypernetwork setup to work worse for a given memory budget than the above discussed other options. An ablation experiment could address these concerns and provide evidence for eventual benefits of the proposed setups. - For evaluation, the method is trained per category, which is a more restrictive setup than most methods are evaluated on. While there is no explicit explanation for this choice of setup, I suspect that otherwise considerably larger MLPs would be required for predicting sufficiently accurate shapes. Predicting the weights of those larger MLPs may be much harder with the hypernetwork or may not work well at all. Furthermore, only 3 of the 55 ShapeNet categories are considered in the experiments. Is that because these are the largest in ShapeNet and all other categories do not provide sufficiently many shapes for learning a decent model? The experiment in Table 2 suggests that the method should be well suited for decent results especially if few models are available. Previous work for single-view 3d reconstruction has commonly evaluated methods trained on all or at least 13 categories jointly. I would thus very much appreciate experiments for more categories and ideally including joint training.

Correctness: The claims and methods appear correct.

Clarity: The paper is well written and easy to follow.

Relation to Prior Work: Prior related work has been sufficiently discussed.

Reproducibility: Yes

Additional Feedback: Reading other reviewer's comments, the paper again, and comparing to the original DVR paper, I have become more concerned about the DVR comparison. The results shown for DVR in this paper are terrible and do not correspond at all to results in the original DVR paper, which is especially concerning as originally DVR was trained on more ShapeNet categories. The reason seems to be the carefully chosen setup for which multi-view consistency for DVR is removed. While the setup is presented as training with single views, up to 24 views of a shape are used during the ShapeNet training nevertheless. I nevertheless think that the paper proposes some interesting ideas, which to me lift it slightly above the acceptance threshold.


Review 4

Summary and Contributions: This submission proposes a method for 3D-unsupervised single-view 3D reconstruction. Unlike previous methods which relied on “multiview” training setting, this work proposes the method that could work on “multi-view” training setting (without view/instance association) and “single-view” training setting. The techniques that differ from previous methods are that the paper : 1. observes the relationship between 2D-SDF and 3D-SDF (the proposition on line 116), and uses it better to supervise SDF outside the visual hull. The loss can be regarded as to enforce an “order” of signed distance instead of only a binary supervision. 2. incorporate an additional network (h) to learn to sample along the ray (potentially more efficiently than pre-design procedures as in NeRF or DVR). A loss function is proposed to enforce those samples before/after hitting the surface to have more distinctive SDF values. Empirically, the paper presents better results in single-view 3D reconstruction, trained under the single-view setting. Further the work also shows its performance on more realistic dataset PASCAL 3D+.

Strengths: The paper addresses an important problem that in the single-view 3D reconstruction method, existing methods tend to rely on the multiview training setting (multi-view images and poses are available for 3D instances). The single view setting or multiview view setting (without instance correspondence) would potentially be very important, as it’s harder to capture and annotate multi-view setting datasets. The design of 2D-SDF loss (L_{SDF}) appears to provide stronger supervision signals, compared to binary occupancy loss. The ray marching network (network h) could potentially accelerate the ray marching process, which in turns could accelerate train/inference speed. Well-written paper and good presentation.

Weaknesses: 1. The work presents good results (qualitative and quantitative) compared to existing methods SoftRas, DVR and CMR. However, I still found it hard to understand which components contribute to the improvement. The underlying difficulty of the single-view training setting lies in the ambiguity of shape and texture: it could be any camera-aligned surface, where the textures are projected onto the surface. The multi-view training (without view/instance association) is similar to this case, with a bit more information in the dataset that there are some same instances (but not explicitly provided to the network) The 2D-SDF loss enforces the learnt SDF outside the visual hull to have an “order”: at least the value from the processed silhouette image (2D-SDF). It doesn’t seem to enforce within the visual-hull volume, therefore the ambiguity still is there. The ray marching network (h) potentially learns to sample the ray in fewer samples; that’s good. The ray loss makes sense to make samples from two sides of the surface to be distinctive to the other side. However, this relies on the h network learnt to sample at the correct depth. In another word, the two networks are collaborating to find a good surface, where they can achieve the agreement at many solutions. Since we don’t have multiview supervision, this is still ambiguous. RGB loss might not help the situation. You can imagine you can print the image on a flat surface. That would still satisfy the loss. The Eikonal regularizer might help in a more “conservative” prediction, as other papers also explore Laplacian loss and flatness loss (e.g. SoftRas). Maybe more explanation would be helpful for the audience to know what has helped the training. 2. The direct comparison case: DVR is also an implicit function based method. The proposed method has a significant advantage over DVR. What might the reason be? To my understanding, DVR utilizes the binary occupancy (SDF-SRN uses the 2D-SDF loss), DVR also did coarse-to-fine ray sampling (uniform sampling then solving for intersection depth by iterative secant method). SDF-SRN learns a network to perform ray sampling: I expect it to be faster, but would it be helpful in learning correct shape without multi-view info? DVR didn’t apply position encoding or Eikonal regularizer. Would these be important factors? Maybe some explanation would be great. 3. Mentioned in 1, there is possibly ambiguity between 3D sdf function f(.) and ray marching function h(.). Imaging network h has recognized the surface as a flat surface, then the f function will reinforce this knowledge. What info does the two explore to get correct supervision (with no multiview info)? 4. Qualitative results of SDF-SRN show “repetitive” artifacts (in most results, especially in Fig 4, 5, 6), . Any explanations?

Correctness: Please see the weakness session

Clarity: Mostly yes Well written paper

Relation to Prior Work: Quite complete related work, although I would love to see the comparison / discussion as mentioned in the weakness session

Reproducibility: Yes

Additional Feedback: ** After rebuttal ** Thank you for the very detailed rebuttal. After reading the rebuttal and discussion with other reviewers, personally speaking, I still hold the following concerns: 1. On the ambiguity issue, the authors gave some explanation on learning a "semantic" correspondence. It's a bit ambiguous explanation. In my most supportive interpretation, is that, e.g., the method learns the shape prior of a chair (how a chair would look like under various views), as in many other single-view image-to-3D regression system. But those methods either have direct supervision or multi-view self-supervision (results at least better than traditional visual hull). But as mentioned in weakness 1 in my original review, to my understanding, no supervising signal is available to shape the chair in correct shape. To my speculation, it might be the dataset setting (shapenet CAD objects with minimal texture, single category, limited camera views), or network weights are nicely initialized (as in Deep Image Prior, CVPR 18 and Deep Geometric Prior for Surface Reconstruction, CVPR 19). The authors mentioned CMR, but CMR also utilizes keypoint annotations, which is an indirect way to provided cross-view association. More discussion would be crucial to convince the reviewers and future audiences. Please also add the details of network initialization (image encoder and the LSTM). 2. Why the proposed method does so much better than DVR (quite similar loss function design). The authors rule of the possibility of influence of positional encoding and eikonal term. The LSTM ray-marching function seems to be stronger than "naive" ray probing, but the proposed method has a "duplicated" design of f() and h() (i.e. f() also does the job to tell if a point is on surface, and h() specializes on searching on a ray), while DVR marches in ray on by guidance of f(). If f() and h() are learned at the same time (so we can regard it as one black box), why it does so much better another black box of f() alone? The authors indeed pointed out binary occupancy could play an important role. In that sense, would DVR be much better if trained with L_SDF and L_ray? Here some additional ablation and discussion would be important. Also please confirm if DVR is trained under exactly the same setting as the proposed method. For the two reasons above, I'm still leaning to reject. Considering this is a move for single-view training and partially-addressed rebuttal, I would slightly raise the rating to 5: marginally below threshold.

[Author Response · NeurIPS 2020]

We thank the reviewers for their valuable comments and suggestions. We are excited that the reviewers identified the
importance of the problem (R3,R4), appreciated the novelty and technical contributions (R2,R3,R4), acknowledged our
superior experimental results (R2,R3,R4), and found the paper professional and well-written (R2,R3,R4). We believe
SDF-SRN takes a significant step towards real-world 3D object reconstruction that can be learned more practically from
static image datasets. R4 affirmed all the merits above, yet thought our work lacked a few explanations (clarified below).
Unfortunately, R1 misunderstood the supervision settings (L223), which led to ill-founded doubts of our contributions
and credibility of our experiments. We address all raised concerns in the following. In addition, we try our best to
resolve the misunderstandings and sincerely hope R4 will reconsider the rating and R1 will re-evaluate at an entirety.

(**R1**) **Misunderstandings.** "Knowledge of CAD model correspondence" refers to the assumption that one knows which
images were created from the same CAD model (L217-224), so one could supervise with effectively all the associated
viewpoints for an input image. This assumption was utilized in the original DVR and SoftRas, but *not* here in the more
practical setup of *training* with *individual* images. It does *not* refer to camera poses, and we have *never* claimed that no
viewpoint information is needed; on the contrary, we have stated they *are* part of the required data (L183,189).

(**R1**) **Overclaimed/Incorrect contributions.** We strongly disagree. Previous works [14,26,27,33] can infer 3D shapes
from a single image, but they must be *trained* with *multiple views*. None of them have shown the ability of *learning*
(not just *inferring*) implicit 3D shapes from single-view images. SDF-SRN can be *trained* with *single-view* supervision
(L10,41,57) concurred by R2, R3, and R4, and we have provided strong results. We urge R1 to recognize the distinction.

(**R1**) **Validity of experiments.** We trained/tested on the same data as DVR, and all methods were run using the released
implementations from the authors. DVR's different performance from the original is attributed to the modified setup
(see above). SRN itself is *not* a 3D reconstruction method and does not produce 3D shapes (L82-87). Other baselines
either did not release source code [26,27] or provided incomplete code [14] at the time of submission, preventing us
from faithfully reproducing results. We chose to compare mainly against DVR for its reproducibility in a fairer setup.

(**R1**,**R4**) **Resolving ambiguities.** Although correspondences across instances are unavailable at the *pixel* level in
single-view training, they still exist at the *semantic* level. CMR has shown initial success [15] in this regard with meshes.
Here, SDF-SRN learns implicit features that best explains the object semantics within the category (L232-236). In turn,
ray-marching discovers and associates implicit *semantic* correspondences in 3D, such that the ray-marched surfaces are
semantically interpretable across all objects. Therefore, shape/depth ambiguities *can* be resolved (albeit not 3D-perfect)
by learning to recover the *appearance* (with $\mathcal{L}_{RGB}$), a classical but important cue for disambiguating 3D geometry. In
SDF-SRN, $f$ (and thus $\mathcal{S}$) would be guided by $g$ and $h$ in ray-marching while also explicitly optimized with $\mathcal{L}_{SDF}$. The
eikonal term has little to do with this mechanism. We hope this clarifies and will include discussions in the final version.

(**R4**) **DVR's drawbacks.** DVR learns by encouraging *binary occupancy* randomly along the rays within the silhouettes
and vice versa, so it relies on other views to "carve" the same shape. Without view-instance association, DVR would
wrongly encourage occupancy at self-occluded regions (L232-236,280-283). SDF-SRN does not rely on such sampling-
based loss, but rather on learning to associate semantic correspondences within category (discussed above). Positional
encoding is unrelated here, and the eikonal term is specific to SDFs. We thank R4 and will clarify in the final version.

(**R4**) **Repetitive artifacts.** We observed that these patterns came from the positional encoding component [31], which
encodes input 3D points with periodic sinusoidal functions. We leave investigation on artifact reduction to future work.

(**R3**) **Robustness.** We thank R3 for the great suggestion. As requested, we tried PASCAL3D+ using *estimated* cameras
from keypoints (L287) and found little performance degradation. We focus on ideal cameras/silhouettes in this work as
the problem is very challenging already, but we agree robustness to such noises is definitely a valuable future direction.

(**R3**) **More categories?** We focused on airplanes, cars, and chairs following [15,27,40,42] as they are the most common.
We thank and agree with R3 that evaluating on more/general categories will add to completeness of our experiments.
We do believe our emphasis on chairs speaks well to SDF-SRN's versatility since chairs exhibit high shape variations.

(**R3**) **Why hypernetworks?** Conditioning inputs on latent code would in fact be *more* costly for ray-marching, since
all unrolled iterations would directly depend on the encoder, resulting in slower backpropagation and higher memory
footprint. Hypernetworks only need one forward/backward pass, allowing much cheaper training (empirically validated).

(**R2**) **Silhouettes.** DVR *does* need silhouettes for the occupancy/freespace losses, so we believe the comparison is fair.
Regarding practicality, one could also obtain object silhouettes in real images from off-the-shelf instance segmentation
methods (*e.g.* Mask R-CNN), which was also originally utilized in CMR [15] (please also see response to R3 on noises).

(**R2**) **Viewpoints.** The single views of ShapeNet were randomly selected from those in L210. The improvement under
the multi-view setup is mainly attributed to the increase of training data, since images were always treated individually.
We did observe slightly degraded results at peculiar viewpoints (*e.g.* wings might shrink at an airplane's side view).

(**R2**) **Runtime.** A batched forward pass of $\mathcal{E}$ to infer the shape parameters $\theta$ takes ~10 ms, and rendering takes ~50 ms.
We chose $N = 10$ steps following SRN [38] and did not observe improvements with more. Evaluating the quality at
intermediate steps is inapplicable since the surfaces only intersect with the rays between the last two steps (L160).

[Meta-Review · NeurIPS 2020]

Two knowledgeable referees supported accept and two knowledgeable referees supported reject. Lack of clarity + emphasis in the writing of the paper had created doubts in reviewers' minds about how and why the method performs as well as it does. The reviewers also had questions about the performance of baselines (DVR in particular). R2 has misunderstood the paper in at least 2 places and despite the author's rebuttal pointing them to their misunderstandings, they have failed to adjust their judgement. I therefore disregard their review. I also believe some of the other reviewers' concerns about the method's surprisingly good performance have been and can be addressed by the authors in revisions. Given the other reviewer comments, and my own reading of he paper, I believe the paper should be accepted as a poster. The paper addresses a significant problem, proposes a non-trivial novel solution, is well written and it is SOTA. I highly encourage the authors to address these issues to the fullest extent possible before publication.